# Dissemination of 2014 dual antiplatelet therapy (DAPT) trial results: a systematic review of scholarly and media attention over 7 months

Melissa K Sharp,[1,2,3] Romana Haneef,[2,3,4] Philippe Ravaud,[1,2,3,4,5] Isabelle Boutron[2,3,4,5]

MKS and RH contributed equally.

[1]Mailman School of Public Health, Columbia University, New York, USA
[2]METHODS Team, Center of Research in Epidemiology and Statistics Sorbonne Paris Cité INSERM UMR 1153, Paris, France
[3]University of Paris Descartes, Paris, France
[4]Centre d'épidémiologie clinique, Hôpital Hôtel Dieu, Assistance Publique des Hôpitaux de Paris, Paris, France
[5]Cochrane France, Paris, France

**Correspondence to**
Professor Isabelle Boutron; isabelle.boutron@aphp.fr

## ABSTRACT

**Objective** To explore how the results from the 2014 dual antiplatelet therapy (DAPT) trial were disseminated to the scientific community and online media.

**Design** A a systematic review of scholarly and public attention surrounding the DAPT study.

**Settings** Data were collected from the ISI Web of Knowledge, Google Scholar, PubMed Commons, EurekAlert, the DAPT study website (www.daptstudy.org) and the *New England Journal of Medicine* website (for scholarly attention) and Altmetric Explorer, Snap Bird, YouTube (for public attention) citing DAPT study results appearing from 16 November 2014 to 10 June 2015.

**Participants** No participants were involved in this study.

**Main outcome measure** Proportion of contents highlighting the increased risk of mortality and critical to the author's interpretation of the results.

**Results** We identified 425 items reported by seven sources; 164 (39%) disseminated the authors' interpretation via an electronic link or a reference, with no additional text. Among 81 items (19 %), the message favoured prolonged treatment and consequently overstated the article conclusions. Among 119 items (28 %), the text was uncertain about the benefit of prolonged treatment but was reported with no or inappropriate mention of increased risk of mortality. Only 34 items (8 %) were uncertain about the benefit of prolonged treatment and mentioned increased risk of mortality. In all, 27 items (6 %) did not favour prolonged treatment, and only 12 of these (3 %) clearly raised some concerns about the reporting of increased risk of death.

**Conclusion** Dissemination of the DAPT study results to the scientific community and on different media sources rarely criticised the interpretation of the study results.

## INTRODUCTION

The development of optimal coronary stent replacement has progressed rapidly over recent years.[1] In the USA, almost 700 000 stents are placed every year and there is an increasing trend for its use in Europe.[2] Dual antiplatelet therapy (DAPT) (ie, P2Y12-receptor inhibitor combined with aspirin) is recommended after placement

of coronary stents to prevent thrombotic complications.[3] The optimal duration of DAPT has been debated.[4–8]

### Strengths and limitations of this study

▶ Our method involved a broad search strategy, ensured to capture an extensive and representative sample of contents citing the 2014 DAPT trial for both scholarly and public attention.
▶ Our systematic approach to analyse the text of contents provides a comprehensive overview of dissemination of the study results.
▶ This study focused on only a specific trial publication and results are not generalisable to other studies.

In December 2014, the Harvard Clinical Research Institute (HCRI) released the results of the DAPT study, the largest international randomised controlled trial to date.[9] The trial aimed to determine the benefits and risks of continuing DAPT beyond 1 year after placement of a coronary stent.[9] A total of 9961 adult patients were randomly assigned to continue thienopyridine treatment or to receive a placebo for 30 months. Continued therapy reduced the rate of stent thrombosis (0.4% vs 1.4%; p<0.001) and major adverse cardiovascular and cerebrovascular events (MACCEs) (2.1% vs 4.1%; p<0.001), with an expected increase in the rate of moderate or severe bleeding (2.5% vs 1.6%; p=0.001).[9] However, continued therapy was also associated with an increase of 36% in all-cause mortality (2.0% vs 1.5%; HR 1.36; (95% CI 1.00 to 1.85; p=0.05).

The results of the DAPT study were published in the *New England Journal of Medicine* (*NEJM*)[9] after their presentation at the American Health Association Conference, in November 2014. However, the reporting of the results raised some concerns.[10 11] Particularly, the abstract

conclusions did not mention the increased risk of mortality. Furthermore, the discussion included explanations based on post hoc analyses to clear the role of prolonged thienopyridine treatment in this increased risk of mortality. For this purpose, the authors had split the analysis by cause of death, which was not powered to show a statistically significant difference. They focused on the increase in cancer-related death (0.62% vs 0.28%, p=0.02). The results were interpreted as being related to an imbalance at baseline in patients with a history of cancer before enrolment (9.8% vs 9.5%). To confirm, the authors performed a post hoc analysis excluding all deaths that could be related to cancer diagnosed before enrolment. Consequently, the results became statistically non-significant (0.50% vs 0.28%, p=0.11). This post hoc exclusion of patients with an event is questionable.

We aimed to explore how the authors' interpretation of results from the DAPT trial was disseminated to the scientific community and online media and to assess whether this interpretation was criticised or not.

## METHODS

We performed a systematic review of scholarly and public attention surrounding the DAPT study.

### Identification of scholarly and public attention surrounding the DAPT study

#### Scholarly attention

On June 2015, we searched the following electronic databases to identify responses to the DAPT study: ISI Web of Knowledge, Google Scholar and PubMed Commons. We also searched the comments and citing articles on the *NEJM* website for the original article.[9]

### Public attention

We searched Altmetric Explorer[12–15] to identify all online attention (news, blogs, Twitter, Facebook, Google+, Mendeley, CiteULike) given to the DAPT study. Each identified social media source was then systematically evaluated to determine whether other posts were not captured by Altmetric Explorer. In addition, each original tweet was reviewed to find retweets, replies and favourites. Since Altmetric.com captures only tweets attached to the Digital Object Identifier (DOI) of the original DAPT article, we also used snapbird.org, a search engine that can search an individual Twitter account by using the *NEJM*'s Twitter account and the search terms 'DAPT' and 'dual antiplatelet therapy'. We also searched EurekAlert! (a free online database for science press releases, www.eurekalert.org) for press releases dedicated to the DAPT study; YouTube (search terms 'DAPT' and 'dual antiplatelet therapy'); and pages dedicated to patients, clinicians and media at the DAPT study website (http://www.daptstudy.org).

### Eligibility criteria

Two researchers (MS and RH) screened all items retrieved and selected all English-language items that cited the DAPT study and were released from 16 November 2014 to 10 June 2015. Any disagreements were resolved by discussion to reach consensus.

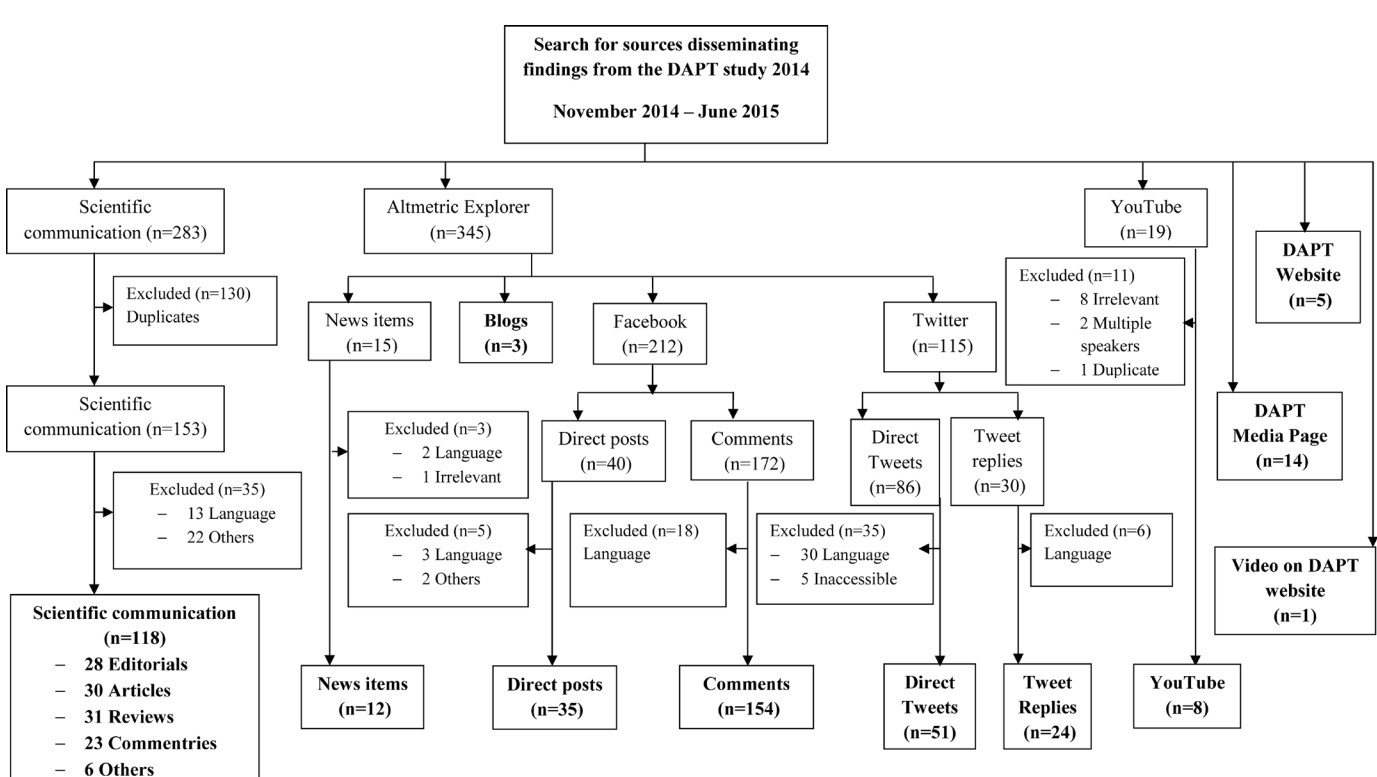

**Figure 1** Flow diagram of identified scholarly and public attention surrounding the dual antiplatelet therapy (DAPT) study.

### Content of scholarly and public attention surrounding the DAPT study

Two researchers (MS and RH) read the items from each source independently and evaluated them by using a preliminarily tested extraction form. Disagreements were resolved by discussion to reach consensus. If needed, a third researcher (IB) appraised the content.

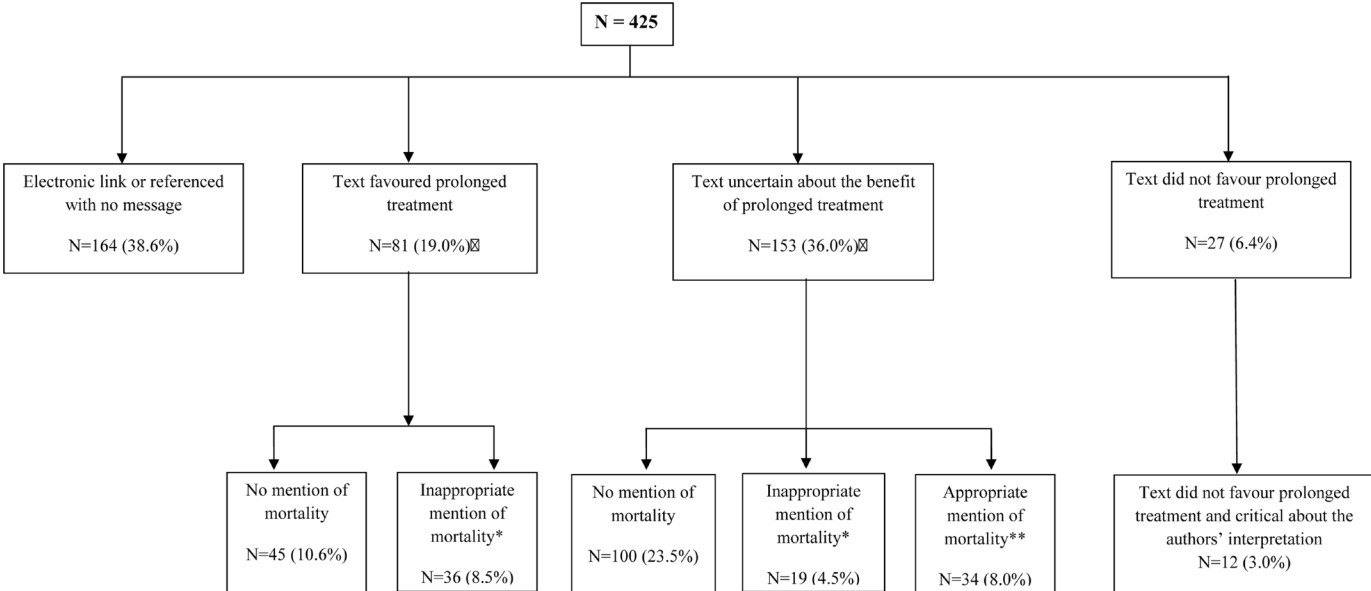

**Figure 2** Content of scholarly and public attention surrounding the dual antiplatelet therapy study (n=425).

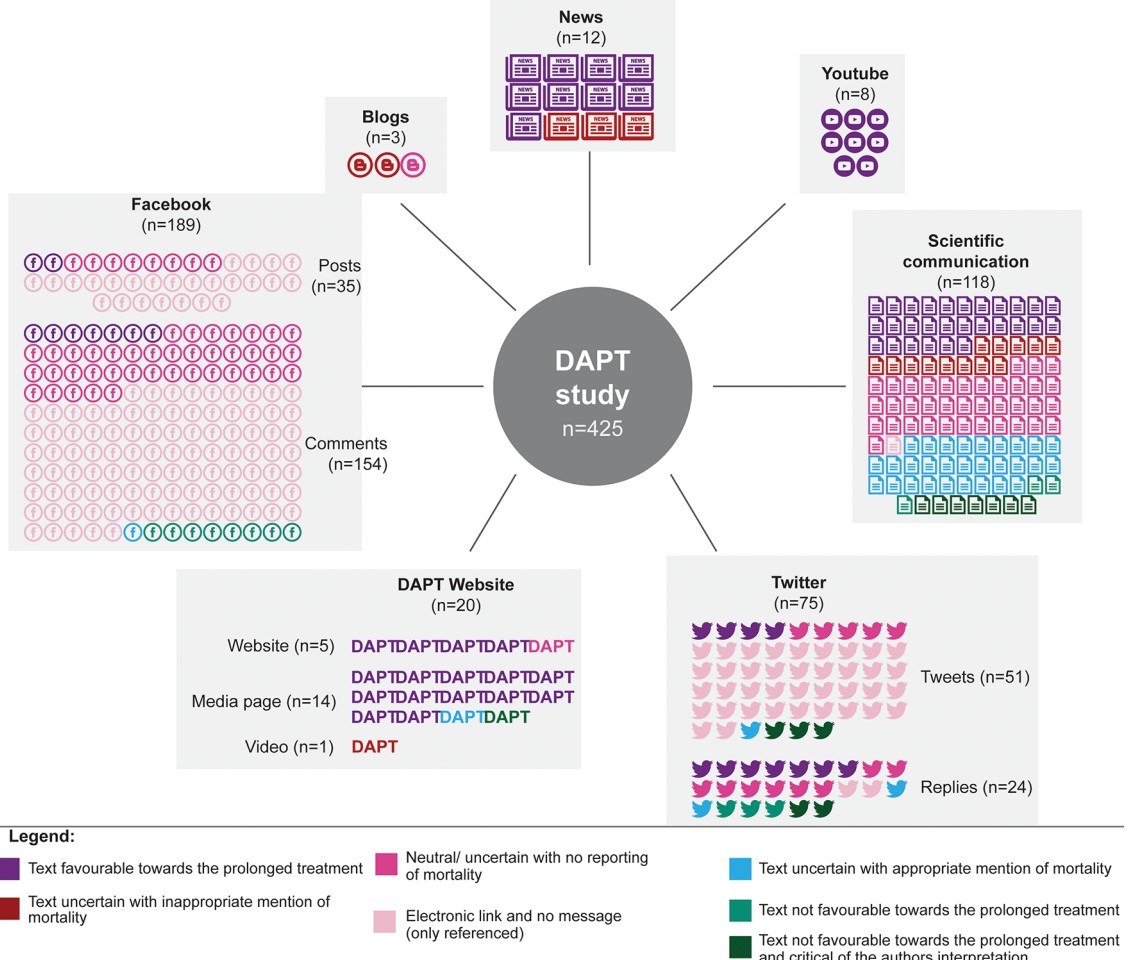

**Figure 3** Content of scholarly and public attention surrounding the dual antiplatelet therapy (DAPT) study by source (n=425).

We determined whether the source consisted of a reference or a link to the *NEJM* article reporting the DAPT study only or was a text commenting on the DAPT study. For a text commenting on the DAPT study, we checked whether the original study authors were involved in writing the text or not. Our main outcome of interest was the proportion of contents highlighting the increased risk of mortality and critical to the author's interpretation of the results. We determined whether

► The primary efficacy outcomes (ie, stent thrombosis and MACCE) were reported.
► The safety outcomes related to moderate or severe bleeding were reported.
► The increased risk of mortality with prolonged treatment was reported.
► The authors' explanation clearing the responsibility of prolonged treatment in the increased risk of mortality was reported or criticised.
► The content of the text was (1) favouring the prolonged treatment and consequently overstating the article conclusion, (2) uncertain about the benefit of the prolonged treatment (ie, statement of both the beneficial effect, and increased risk of bleeding, text ending with a question mark, use of 'may or might' or reporting that the study needs further research) or (3) not favouring the prolonged treatment.[16]

Overall, we classified the sources based on the text of contents as follows:

► text favouring the prolonged treatment
► text uncertain (about the benefit of prolonged treatment) with inappropriate mention of mortality
► text neutral/uncertain (about the benefit of prolonged treatment) with no mention of mortality
► electronic link or referenced with no message
► text uncertain (about the benefit of prolonged treatment) with appropriate mention of mortality
► text not favouring the prolonged treatment
► text not favouring the prolonged treatment and critical of the authors' interpretation.

### Statistical analysis

We calculated frequencies and percentages for qualitative variables and median (IQR) for quantitative variables.

### RESULTS
### Identification of scholarly and public attention surrounding the DAPT study

From all sources, we selected and appraised 425 items: 118 scientific communications, 12 news items, three blogs, 189 Facebook posts or comments, 75 tweets or replies, eight videos on YouTube, 14 DAPT media pages, five DAPT website pages and one video on the DAPT website (figure 1). The original study authors were directly involved in 35 items. Details of 118

scientific communications are given in Appendix 1 in the online supplementary file 1 .

### Reporting of the content

The texts of contents are described in figure 2 (overall) and figure 3 (by source). Overall, 164 items (39%) involved disseminating the authors' reporting and interpretation via an electronic link (n=151, 36%) or reference (n=13; 3%), with no additional text or message. Among 81 items (19%), the message favoured the prolonged treatment and therefore overstated the article conclusions. For example, the DAPT study website dedicated to patients reported that '*It is important that patients who currently take a thienopyridine anti-clotting medication (clopidogrel or prasugrel) do not stop taking their medication. […] The benefits of continuing dual antiplatelet therapy for 1 year, according to current guidelines, far outweigh the risks*'. Among 153 items (36%), the text was uncertain about the benefit of prolonged treatment but was reported with no mention of the increased risk of mortality (n=100, 24%) or the authors' explanation clearing the responsibility of prolonged treatment (n=19; 4.5%). Overall, 34 items (8%) were uncertain about the benefit of prolonged treatment but mentioned the increased risk of mortality. Only 27 (6%) did not favour prolonged treatment and only 12 of these (3%) clearly raised some concerns about the reporting of the increased risk of death. Further information on items by source is in Appendix 2 in the online supplementary file 2.

Overall, 136 items (32%) reported efficacy outcomes (ie, stent thrombosis and MACCEs), 127 (30%) safety outcomes and 113 (27%) both efficacy and safety outcomes.

A total of 100 items (24%) did not mention mortality, but when mortality was mentioned, in 19 items (5%), it was reported with the authors' justification for prolonged treatment.

### DISCUSSION

We describe the dissemination of the 2014 DAPT study findings in scientific community and to the public via different sources such as news, blogs and social media. Our assessment of 425 items disseminating the DAPT study results showed that only 8% of the items mentioned some uncertainty about the benefit of prolonged treatment and included a mention of the increased risk of mortality. Furthermore, only 12 items (3%) clearly raised some concerns about the reporting of the increased risk of death. This study adds to the burgeoning literature on the biased dissemination of research results. Previous studies have focused on publication bias,[17] selective reporting of outcomes,[17–22] and spin.[19 23 24]

However, this is the first study to our knowledge to focus on both scholarly and public dissemination of

study results. Our study highlighted an unmet need of scientific communication in the media, whose importance in dissemination of scientific data is becoming increasingly relevant. These findings could be helpful for the entire community for better understanding how scientific knowledge is disseminated.

Our approach involved a broad search strategy and multiple search engines, which ensured the capture of an extensive and representative sample of contents discussing the DAPT study results. Each social media item from Altmetric was systematically reviewed for additional content that may have been missed, and several different search engines were used. We captu red items that were published over the course of many months, which highlighted the perpetuation and continuation of the dissemination of the authors' interpretations. The inclusion period for sources seemed to be more than sufficient because tweets linked to scientific articles have been shown to taper off well before our cut-off point (7 months).[25] In addition, two independent researchers assessed each source by using a standardised data extraction form and disagreements were resolved by consensus.

However, our study has some limitations. First, this study focused on only a specific trial publication and results are not generalisable to other studies. However, the article we focused on was among the top five of all research outputs and within the 99th percentile of articles on Altmetric. Second, the data extraction involved some subjectivity; however, we tried to address this by using a standardised data extraction form and independent assessment as well as consensus among two researchers. Third, despite our best efforts, we cannot ensure that our search strategy was all-encompassing because of the breadth of social media. Finally, we did not explore the balance between efficacy and safety outcomes with DAPT treatment.

Our aim was not to resolve the controversy about DAPT duration and this debate is still ongoing. The Optimal Duration of Dual Antiplatelet Therapy After Drug-eluting Stent Implantation (OPTIDUAL) trial did not find an increased risk of death with the prolonged treatment; on the contrary, the risk of death was lower with the prolonged treatment.[26] Several meta-analyses found conflicting results.[4 5 8 27 28] The researchers involved in the DAPT trial concluded in a meta-analysis published in *The Lancet* that prolonged DAPT duration was not associated with a difference in risk of all-cause mortality.[29] Three meta-analyses, published later by different teams, showed prolonged DAPT associated with increased risk of all-cause mortality.[4 5 8] More recently, other meta-analyses did not find a statistically significant increase in all-cause mortality.[27 28] Most of these meta-analyses warranted further research with extended DAPT.

However, these results are difficult to interpret because of different definitions of short (1, 3, 6 or 12 months) and extended (6, 12, 24 or >24 months)

durations, which varied across studies. Furthermore, different durations of follow-up and types of stents could also influence the results.

## CONCLUSIONS

Dissemination of the DAPT study results to the scientific community and on different media sources rarely criticised the interpretation of the study results.

**Acknowledgements** We thank Elise Diard for help in creating figure 2. We acknowledge the support from Altmetric for free access to 'Altmetric Explorer'. We acknowledge the assistance in English language proofreading by Laura Smales (BioMedEditing, Toronto, Canada). Isabelle Boutron and Philippe Ravaud submitted a letter to the NEJM following the publication of the DAPT study to highlight the inadequate reporting in the abstract conclusions, but the letter was rejected.

**Contributors** Study conception, design, selection of contents and data extraction: MS, RH. Study conception and design: MS, RH, IB. Selection of contents, data extraction: MS, RH. Analysis of data and interpretation of results: RH, PR, IB. Contributed to the writing of the manuscript: MS, RH, PR, IB. All authors read and approved the final manuscript.

**Competing interests** 1.Isabelle Boutron is deputy director of the French Equator centre and coordinator of the MiRoR (Methods in Research on Research) project funded by the European Union's Horizon 2020 research and innovation programme under the Marie Sklodowska-Curie grant agreement No 676207. 2.Phlippe Ravaud is member of the steering committee of the Equator network and director of the French Equator centre. 3.Melissa Sharp is fellow in the MiRoR (Methods in Research on Research Project) funded by the European Union's Horizon 2020 research and innovation programme under the Marie Sklodowska-Curie grant agreement No 676207.

**Provenance and peer review** Not commissioned; externally peer reviewed.

**Data sharing statement** All relevant data are included in this manuscript. Details of text content are available upon request for academic researchers.

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
