## [Reviewer comments · BMJ Open]

ARTICLE DETAILS

TITLE (PROVISIONAL)	Dissemination of 2014 Dual Anti-platelet Therapy (DAPT) trial results: A systematic review of scholarly and media attention over 7 months
AUTHORS	Sharp, Melissa K; HANEEF, Romana; Ravaud, Philippe; Boutron, Isabelle

VERSION 1 - REVIEW

REVIEWER	Victor Serebruany Johns Hopkins, USA
REVIEW RETURNED	19-Oct-2016

GENERAL COMMENTS	Great paper! Bright and solid. What is almost completely missing are references. The paper states 118 scientific contributions - they should be listed in the References. Not fair to omit them since the idea that DAPT was (and still is) hiding cancer risks and avoid discussing cancer deaths is not new. Other than that I am totally supportive of this work.
--

REVIEWER	Bernard Cheung University of Hong Kong
REVIEW RETURNED	17-Dec-2016

GENERAL COMMENTS	This paper is thought-provoking but in the last analysis, not particularly informative, helpful or constructive about the current controversies regarding the DAPT duration. It discussed the impact of the dissemination of the DAPT trial. The results criticized that the scholarly and public attention were heavily misled by the "questionable interpretation" from the DAPT trial. It is true that the interpretation of the DAPT trial focused more on the benefits of extending DAPT duration, and the increased risk of bleeding. Few papers in the literature commented on the relatively weak explanation of the increased risk of mortality, which is the strength of this paper, in redressing the balance. The increased risk of all-cause mortality in the extended DAPT group in the DAPT trial is quite borderline ($P=0.05$). In the later OPTIDUAL trial, the all-cause mortality did not differ significantly between 12 vs. 30 months' DAPT. Thus this submitted paper is also suffering from the fault it criticizes, that is, overstating the "misleading and inappropriate" impact of the DAPT trial. The manuscript is not quite up to date. If the authors start to read the
--

	more recent literature as well as meta-analyses, the conclusions may perhaps be different. There are some places in the manuscript where British spelling has not been followed.
--	--

REVIEWER	Bundhun, Pravesh Department of Cardiovascular Diseases, The First Affiliated hospital of Guangxi Medical University China
REVIEW RETURNED	09-May-2017

GENERAL COMMENTS	This is an interesting topic which aimed to focus on the DAPT trial and its negative points. However, because this study is not generalized, I do not think it will be of significant importance. I agree with one strong point of this current study: Many studies did not comment about mortality rate. A recently published meta-analysis based on 15 randomized controlled trials showed prolonged DAPT use to be associated with lower mortality rate when compared to short duration DAPT use. This was a generalized result based on data obtained from a pooled analysis of patients. Why did the author only focus on the DAPT trial? Any specific reason?
---

REVIEWER	Marco Valgimigli Inselspital, Bern, Switerlamd
REVIEW RETURNED	28-May-2017

GENERAL COMMENTS	In the manuscript entitled "Lack of critical opinion to highlight misleading interpretation of increased risk of mortality in the Dual Anti-platelet Therapy (DAPT) study" Sharp and colleagues conducted a cross-sectional review of scholarly and public media release regarding the DAPT study, which was a large randomized trial comparing in nearly 10,000 patients a prolonged thienopyridine treatment for 18 months with placebo after a standard course of 12 months dual antiplatelet therapy. The trial showed a significant reduction in the risk of the two primary efficacy endpoints of stent thrombosis and major adverse cardiovascular and cerebrovascular events at the cost of an increased risk of clinically relevant bleeding. An additional finding of the DAPT study was a borderline increase in the risk of mortality among patients allocated to prolonged thienopyridine treatment compared with placebo (2.0% vs 1.5%, P=0.05). Such finding draw the attention of the authors that specifically focussed on the public resonance of a possibly higher hazard of mortality associated with prolonged DAPT. The authors should be commended for having painstakingly conducted an original review on multiple platforms in a very exhaustive manner. Moreover, their findings are appropriately illustrated and could be helpful for the entire scientific community in better understanding how scientific knowledge is disseminated. This reviewer has the following minor comments:  - The introduction requires a more balanced and less far-fetched overview of the problem. Indeed, it is important to emphasize that the DAPT study was not powered for mortality and therefore the
--

	higher risk of mortality may be the result of the play of the chance. In addition, the hazard ratio included the null effect and therefore it is not conclusive. Obviously, the study has to be interpreted with the caveat that a prolonged thienopyridine treatment could be associated with a higher risk of death.  - In this context, it is important to highlight that meta-analyses on the topic have provided conflicting results. The authors appropriately quoted the meta-analyses that have been performed on the topic but again it is important to provide a more balanced interpretation of the findings. - Page 5, Lines 17-23. The increased risk of mortality associated with prasugrel remains controversial. A recent systematic review on the topic concluded that there is currently insufficient evidence to suggest that thienopyridine exposure is associated with an increased risk of cancer event rate or mortality (Kotronias RA et al. Drug Saf 2017). Please revise. - The post-hoc analysis of the DAPT study regarding the causes of death in the trial is not discussed in a balanced fashion. The incidence of cancer was similar between the experimental and control arms (0.4% vs. 0.5%, P=0.57). As a consequence, a statement like “instead of raising the hypothesis that prolonged treatment could increase the risk of cancer” appears to be biased. Carcinogenesis is a process that typically takes several years or even decades and from a biological point of view it is quite unlikely that thienopyridine treatment would not only promote cancer formation but also being clinically detectable and relevant in a relatively short period of time. Please revise. - As limitation, I would suggest that the study investigated only one specific finding (mortality), whereas the equipoise between ischemic prevention and bleeding risk has not been formally explored. In other words, the balance between efficacy and safety has not been addressed by the study and this represents an important aspect given the accumulating evidence showing a similar prognostic impact of myocardial infarction and bleeding on the risk of subsequent mortality. - The authors may wish to consider changing the title of the study by adding that the study is a systematic review of scholarly and public media. - In the discussion, the authors may wish to mention that the study in a way uncovers an unmet need in the scientific communication media whose importance in the dissemination of scientific data is becoming more and more relevant.
--	---

VERSION 1 – AUTHOR RESPONSE

Reviewer # 1

Bright and solid.

What is almost completely missing are references. The paper states 118 scientific contributions - they should be listed in the References. Not fair to omit them since the idea that DAPT was (and still is) hiding cancer risks and avoid discussing cancer deaths is not new.

Other than that I am totally supportive of this work.

Answer: Thank you very much for your appreciation and useful inputs.

We have included the 118 references as requested in an appendix 1 and reported in supplementary data as follow (page 13):

“Appendix 1: Detail of 118 scientific communications”

Reviewer # 2

This paper is thought-provoking but in the last analysis, not particularly informative, helpful or constructive about the current controversies regarding the DAPT duration.

It discussed the impact of the dissemination of the DAPT trial. The results criticized that the scholarly and public attention were heavily misled by the “questionable interpretation” from the DAPT trial. It is true that the interpretation of the DAPT trial focused more on the benefits of extending DAPT duration, and the increased risk of bleeding. Few papers in the literature commented on the relatively weak explanation of the increased risk of mortality, which is the strength of this paper, in redressing the balance.

The increased risk of all-cause mortality in the extended DAPT group in the DAPT trial is quite borderline ($P=0.05$). In the later OPTIDUAL trial, the all-cause mortality did not differ significantly between 12 vs. 30 months' DAPT. Thus this submitted paper is also suffering from the fault it criticizes, that is, overstating the “misleading and inappropriate” impact of the DAPT trial. The manuscript is not quite up to date. If the authors start to read the more recent literature as well as meta-analyses, the conclusions may perhaps be different.

There are some places in the manuscript where British spelling has not been followed.

Answer:

Thank you your comments.

This paper is not informative about the current controversies regarding the DAPT duration because our aim was not at all to resolve this controversy, but to explore the dissemination of the questionable presentation and interpretation of the DAPT trial and to assess whether it was criticized or counterbalanced by other researchers. This has been clarified in the text.

We agree that we should not overstate our point and that some references were missing, probably because there was a delay of 13 months since the submission of this manuscript. We consequently revised the manuscript to avoid overstatement and used a balanced language. We also now reference more recent publications on this topic.

However, the results showed an increase of 36% in risk of all-cause mortality ($HR= 1.36$ [95% CI 1 to 1.85]). We strongly believe that such effect should not be ignored only because the p-value was “borderline,” particularly when the outcome is all-cause mortality.

Indeed, according to the recent ASA's statement on p-value, “Scientific conclusions and business or policy decisions should not be based only on whether a p-value passes a specific threshold.”

Finally, the manuscript has been proofread.

Introduction Section (in the end)

“We aimed to explore how the distorted interpretation of results from DAPT trial was disseminated to the scientific community and online media and to assess whether this was criticized or not.”

Discussion section (in the start)

“We described [...]. Our aim was not to resolve the controversies about DAPT duration.”

Discussion section (page 12, last paragraph)

“Our aim was not to resolve the controversy about DAPT duration and this debate is still ongoing. The OPTIDUAL trial did not find an increased risk of death with the prolonged treatment; on the contrary, the risk of death was lower with the prolonged treatment (Helft et al 2016). Several meta-analyses found conflicting results (Palmerin et al 2015, Spencer et al 2015, Navares et al 2015, Basaraba et al 2017, D’ascenzo et al 2016). The researchers involved in the DAPT trial concluded in a meta-analysis published in The Lancet that prolonged DAPT duration was not associated with a difference in risk of all-cause mortality (Elmariah et al 2014). Three meta-analyses, published later by different teams, showed that prolonged DAPT was associated with increased risk of all-cause mortality (Spencer et al 2015, Navares et al 2015, Palmerin et al 2015). More recently, other meta-analyses did not find a statistically significant increase in all-cause mortality, although the study lacked power and favoured short term DAPT (Basaraba et al 2017, D’ascenzo et al 2016). Most of these meta-analyses warranted further research with extended DAPT.

However, these results are difficult to interpret because of the different definitions of short (1, 3, 6, or 12 months) and extended (6, 12, 24 or > 24 months) durations, which varied across studies. Furthermore, different durations of follow-up and types of stents could also influence the results.”

Conclusion

Abstract’s conclusion (page 2)

“The amount of content criticizing the interpretation of the DAPT study results was limited.”

Full-text conclusion (page 13)

“Dissemination of the DAPT study results to the scientific community and on different media sources rarely criticized the interpretation of the study results.”

Reviewer # 3

This is an interesting topic which aimed to focus on the DAPT trial and its negative points. However, because this study is not generalized, I do not think it will be of significant importance. I agree with one strong point of this current study: Many studies did not comment about mortality rate. A recently published meta-analysis based on 15 randomized controlled trials showed prolonged DAPT use to be associated with lower mortality rate when compared to short duration DAPT use. This was a generalized result based on data obtained from a pooled analysis of patients.

Why did the author only focus on the DAPT trial? Any specific reason?

Answer: Thank you for your comment.

We strongly disagree that this study is not of significant importance. Optimal duration of DAPT is important because it is used by millions of patients who receive stents every year all over the world. The content of publications and media can impact researchers, physicians and patient behaviour (Matthews A et al, BMJ 2016). Our manuscript clearly highlights that the questionable reporting and interpretation of the trial was widely disseminated and the message was rarely corrected.

This paper did not aim to resolve the debate on DAPT, and we are fully aware that we do not have a clear answer whether DAPT actually increases mortality or not. However, our results showed that when researchers misreport their trial results, there is no subsequent correction.

We focused on DAPT trial because we read it for a journal club and we were shocked by the interpretation and presentation of the results.

Reviewer # 4

In the manuscript entitled “Lack of critical opinion to highlight misleading interpretation of increased risk of mortality in the Dual Anti-platelet Therapy (DAPT) study” Sharp and colleagues conducted a cross-sectional review of scholarly and public media release regarding the DAPT study, which was a large randomized trial comparing in nearly 10,000 patients a prolonged thienopyridine treatment for 18 months with placebo after a standard course of 12 months dual antiplatelet therapy. The trial showed a significant reduction in the risk of the two primary efficacy endpoints of stent thrombosis and major adverse cardiovascular and cerebrovascular events at the cost of an increased risk of clinically relevant bleeding.

An additional finding of the DAPT study was a borderline increase in the risk of mortality among patients allocated to prolonged thienopyridine treatment compared with placebo (2.0% vs 1.5%, $P=0.05$). Such finding draw the attention of the authors that specifically focused on the public resonance of a possibly higher hazard of mortality associated with prolonged DAPT.

The authors should be commended for having painstakingly conducted an original review on multiple platforms in a very exhaustive manner. Moreover, their findings are appropriately illustrated and could be helpful for the entire scientific community in better understanding how scientific knowledge is disseminated.

This reviewer has the following minor comments:

1- The introduction requires a more balanced and less far-fetched overview of the problem. Indeed, it is important to emphasize that the DAPT study was not powered for mortality and therefore the higher risk of mortality may be the result of the play of the chance. In addition, the hazard ratio included the null effect and therefore it is not conclusive. Obviously, the study has to be interpreted with the caveat that a prolonged thienopyridine treatment could be associated with a higher risk of death.

Answer: Thank you very much for this comment.

We have updated the introduction section in a more balanced language. However, even if the DAPT study was not powered for mortality and the p-value was borderline, we strongly believe that such results should be clearly reported and highlighted. Indeed, according to the recent ASA statement on p-values, “Scientific conclusions and business or policy decisions should not be based only on whether a p-value passes a specific threshold.”

The results showed an increase of 36% in all-cause mortality (HR= 1.36 [95% CI 1 to 1.85]). We strongly believe that such effect should not be ignored only because the p-value was “borderline,” particularly when the outcome is all-cause mortality.

Nevertheless, we agree that we should not overstate our point and we revised the manuscript to avoid overstatement and used a balanced language. We particularly deleted all results related to an increase risk of cancer with DAPT.

2- In this context, it is important to highlight that meta-analyses on the topic have provided conflicting results. The authors appropriately quoted the meta-analyses that have been performed on the topic but again it is important to provide a more balanced interpretation of the findings.

Answer: As requested, we have now revised the paper to provide a more balanced interpretation of the findings.

3- Page 5, Lines 17-23. The increased risk of mortality associated with prasugrel remains controversial. A recent systematic review on the topic concluded that there is currently insufficient evidence to suggest that thienopyridine exposure is associated with an increased risk of cancer event rate or mortality (Kotronias RA et al. Drug Saf 2017). Please revise.

Answer: We agree. At the time of submission (13 months ago), this article was not published. We

have deleted the controversial sentence in the introduction.

4- The post-hoc analysis of the DAPT study regarding the causes of death in the trial is not discussed in a balanced fashion. The incidence of cancer was similar between the experimental and control arms (0.4% vs. 0.5%, $P=0.57$). As a consequence, a statement like “instead of raising the hypothesis that prolonged treatment could increase the risk of cancer” appears to be biased. Carcinogenesis is a process that typically takes several years or even decades and from a biological point of view it is quite unlikely that thienopyridine treatment would not only promote cancer formation but also being clinically detectable and relevant in a relatively short period of time. Please revise.

Answer: We fully agree. We have deleted this sentence and updated the introduction section as follows (page 4 & 5).

“For this purpose, the authors had split the analysis by cause of death, which was not powered to show a statistically significant difference. They focused on the increase in cancer-related death (0.62% vs 0.28%, $p = 0.02$). The results were interpreted as being related to an imbalance at baseline in patients with a history of cancer before enrollment (9.8% vs 9.5%). To confirm this, the authors performed a post-hoc analysis excluding all deaths that could be related to cancer diagnosed before enrolment. Consequently, the results became statistically non-significant (0.50% vs 0.28%), $p = 0.11$). This post-hoc exclusion of patients with an event is a concern.

5- As limitation, I would suggest that the study investigated only one specific finding (mortality), whereas the equipoise between ischemic prevention and bleeding risk has not been formally explored. In other words, the balance between efficacy and safety has not been addressed by the study and this represents an important aspect given the accumulating evidence showing a similar prognostic impact of myocardial infarction and bleeding on the risk of subsequent mortality.

Answer: Thank you for your comment.

We agree and added this as a limitation in the discussion section as follows (page 12, fourth paragraph):

“Finally, we did not explore the balance between efficacy and safety outcomes with DAPT treatment.”

6- The authors may wish to consider changing the title of the study by adding that the study is a systematic review of scholarly and public media.

Answer: Thank you for this comment. We agree and have modified the title by adding the study design as follows:

“Dissemination of 2014 Dual Anti-platelet Therapy (DAPT) trial results: A systematic review of scholarly and public media attention over 7 months”

7- In the discussion, the authors may wish to mention that the study in a way uncovers an unmet need in the scientific communication media whose importance in the dissemination of scientific data is becoming more and more relevant.

Answer: Thank you very much for this useful suggestion. We have mentioned this point in the discussion section (page 11, second paragraph).

“However, this is the first study [...]. Our study highlighted an unmet need of scientific communication in the media, whose importance in dissemination of scientific data is becoming increasingly relevant.

These findings could be helpful for the entire scientific community for better understanding how scientific knowledge is disseminated.”

VERSION 2 – REVIEW

REVIEWER	Bernard Cheung University of Hong Kong Hong Kong
REVIEW RETURNED	05-Jul-2017

GENERAL COMMENTS	Thank you for revising the manuscript. All my comments have been addressed.
---

REVIEWER	Marco Valgimigli Bern University Hospital Switzerland
REVIEW RETURNED	19-Jul-2017

GENERAL COMMENTS	The authors adequately addressed all comment.
---